# Post-Harvest Operations to Generate High-Quality Medicinal Cannabis Products: A Systemic Review

**DOI:** 10.3390/molecules27051719

**Published:** 2022-03-06

**Authors:** Hebah Muhsien Sabiah AL Ubeed, Ronald B. H. Wills, Jayani Chandrapala

**Affiliations:** 1School of Science, College of Sciences, Engineering, Computing Technologies and Health and Medical Sciences, RMIT University, Bundoora, Melbourne, VIC 3083, Australia; jayani.chandrapala@rmit.edu.au; 2School of Environmental and Life Sciences, University of Newcastle, Ourimbah, NSW 2258, Australia; ron.wills@newcastle.edu.au

**Keywords:** *Cannabis*, medicinal *Cannabis*, cannabinoids, drying technology, post-harvest

## Abstract

The traditional *Cannabis* plant as a medicinal crop has been explored for many thousands of years. The *Cannabis* industry is rapidly growing; therefore, optimising drying methods and producing high-quality medical products have been a hot topic in recent years. We systemically analysed the current literature and drew a critical summary of the drying methods implemented thus far to preserve the quality of bioactive compounds from medicinal *Cannabis*. Different drying techniques have been one of the focal points during the post-harvesting operations, as drying preserves these *Cannabis* products with increased shelf life. We followed or even highlighted the most popular methods used. Drying methods have advanced from traditional hot air and oven drying methods to microwave-assisted hot air drying or freeze-drying. In this review, traditional and modern drying technologies are reviewed. Each technology will have different pros and cons of its own. Moreover, this review outlines the quality of the *Cannabis* plant component harvested plays a major role in drying efficiency and preserving the chemical constituents. The emergence of medical *Cannabis,* and cannabinoid research requires optimal post-harvesting processes for different *Cannabis* strains. We proposed the most suitable method for drying medicinal *Cannabis* to produce consistent, reliable and potent medicinal *Cannabis*. In addition, drying temperature, rate of drying, mode and storage conditions after drying influenced the *Cannabis* component retention and quality.

## 1. Introduction

*Cannabis* has been a traditional medicinal herb in central Asia [1], with reports of such usage back to 4700 B.P. in China, India, Persia, Egypt, Greece and Rome [1]. It is now also cultivated and used as a drug crop in at least 172 countries and territories worldwide [2]. It is classified as *Cannabis Sativa*, *C. Indica* and *C. Ruderalis* based on genetics, phenotypic properties and chemical structure [3]. All the classes have medicinal cannabinoids compounds, but in different proportions. For instance, *Cannabis Sativa* has a high level of Cannabidiol (CBD), while *C. Indica* and *Rudelaris* have high and low levels of Δ^9^-tetrahydrocannabinol (THC) respectively [3,4,5].

Aizpurua-Olaizola, et al. [6] listed 554 compounds identified in *Cannabis* plants, including 125 cannabinoids and 198 non-cannabinoid compounds like phenols and flavonoids, terpenes and alkaloids [7,8]. Cannabinoids are the active medicinal constituents against the development of numerous conditions. CBD has therapeutic activity against antipsychotic, anti depressive, anxiolytic, antiepileptic, anti spasticity and anti-inflammatory, stroke and hypoxic-ischemic, spinal cord injury, rheumatoid arthritis and various types of cancer, such as brain, blood, breast, lung, prostate and colon [9,10,11,12,13,14]. However, THC is well-documented having anti-inflammatory effects, including for arthritic and inflammatory conditions [15], Alzheimer’s disease [16], Parkinson’s [17] and diabetes [18]. However, cannabigerol (CBG) and cannabichromene (CBC) have antibacterial and antifungal effects [19], and can act as antidepressants [20]. In contrast, cannabinol (CBN) has a potential effect on insomnia and sleep disorder [21].

Production of cannabinoids in *Cannabis* is mainly derived from cannabigerolic acid (CBGA), or the mother of the *Cannabis* via co-enzyme Olivetolate geranyl transferase such as tetrahydrocannabinolic acid synthase (THCAS) cannabidiolic acid synthase (CBDAS) or cannabichromene acid synthase (CBCAS) [22], to produce tetrahydrocannabinolic acid (THCA), cannabidiolic acid (CBDA) and cannabichromenic acid (CBCA), and oxidation of THCA to produce cannabinolic acid (CBNA) in the resin glands or trichomes [23,24]. Then these naturally cannebinoids acids are converted via decarboxylation, which removes the carboxylic acid functional group from the cannabinoids via drying, heating or combustion to produce CBD, CBC, CBG and oxidation THC to Delta-8-tetrahydrocannabinol (Δ^8^-THC) and CBN [25,26,27] (Figure 1).

Medicinal *Cannabis* has traditionally been used to treat various illnesses using different plant parts [28]. According to Jin, et al. [29], leaves are rich in cannabinoids (1.10–2.10%), terpenoids (0.13–0.28%) and flavonoids (0.34–0.44%). The seed oil is mostly used in Arab and Chinese medicine [30,31]. However, a product derived from seed and its concentration still required further study [32]. In addition, a recent study by Lima, et al. [33] reported no spasmolytic effect, no toxicity effect and reduction in edema formation at all tested doses (12.5, 25, 50 and 100 mg/kg) of aqueous extract of *Cannabis Sativa* roots (CsAqEx) on the airway smooth muscle in mice. In modern medicine, however, *Cannabis* female inflorescence is the most used part of the plant, and contains the highest concentration of different cannabinoids and active terpenes [34]. *C. Sativa* species regulated the cell death in the plant via accumulated cannabinoids in the glands above the leaf [35]. Over 300 years, the functions of these glands have been well known as either attracting pollinators or protecting against pathology. However, *Cannabis* glands contributed additional functions as death capsules [36,37]. These cannabinoids result from secondary metabolism in *Cannabis* [38]. Cannabinoids accumulate where death happens in the plant tissue [39]. Cannabinoid resin causes necrotic and apoptotic cells through DNA degradation mediated by caspase-dependent nuclease and catalysed by nuclease released from mitochondrial during action mitochondrial permeability transition (MPT) [39]. Secretion of these cannabinoids or treating the plant with these cannabinoids will stimulate the defense system during the initial and late stages of the leaf senescence [36,37]. Another significant benefit is to fulfil the requirement for the completed growth cycle and eventually develop the seeds in the medicinal *Cannabis* [36]. In general, senescence happens during all growing cycles, and then ultimately leads to death [40]. This death plays an essential role in diverse physiological actions, including root cap, stomatic embryogenesis, xylogenesis, leaf senescence and defense against microbial pathogens and abiotic stresses [41]. Due to secondary metabolites, changes in the active compounds like THCA and THC led to altered chemical composition [38,42].

However, extending the post-harvest life of medicinal *Cannabis* reduces the rate of metabolism, minimises exposure to post-harvest diseases and limits handling-related damage [43].

Due to decarboxylation, many processes disrupt the chemical composition, such as oxidise or isomerise. [44,45]. Grafström, et al. [46] showed that oxygen increases the decarboxylation of THCA to THC. When inflorescences (buds) are stored under high temperatures. During storage, mostly over 24 h, CBN is formed by the decarboxylation of THC at temperatures over 50 °C [45,47]. However, CBD is at risk of oxidative degradation, and is stable over time [47], since *Cannabis* is a plant-based drug that, unlike cocaine or heroin, requires no plant processing to produce the active constituents in the administered drug [48].

Post-harvest reports optimising the drying techniques and factors to recover bioac-tive compounds such as terpenes, cannabinoids and phenolic compounds from the medicinal marijuana are limited in the literature, so there is a need to understand the post-harvest handling of *Cannabis* that optimise yield and control high-quality medical-grade product. Typical processing of *Cannabis* involves harvesting at the optimal time, drying the harvested product with minimal loss of active constituents, extracting cannabinoids, then storing the product for further process. This review shows that the knowledge for the optimal benefit from the post-harvest process is standardised to produce therapeutically stable *Cannabis* products for patients by a deeper understanding of its chemical space, outlining major phytochemicals in *Cannabis* and reviewing the current drying techniques and optimum methods which have been applied for *Cannabis*-derived phytochemicals. Highly efficient mechanical drying methods are needed to effectively reduce the costs of drying, and phytochemical isolation is crucial for further applications. To date, there are not many papers that discuss the post-harvesting strategies for medicinal *Cannabis* and managing the quality of export *Cannabis*.

## 2. Harvesting of *Cannabis*

There is a significant difference in the potency, quality and content of cannabinoids and terpenes between unripe and ripe buds [26,49,50]. When the bud is ripe, it is the best time to harvest *Cannabis* [50]. Therefore, daily bud inspections and extra time to harvest will feature multiple harvesting sessions to ensure the finest harvest and best quality to process medicinal *Cannabis* [49]. The following section will explain how to determine the time to harvest and the best harvesting technology.

### 2.1. Determining the Time to Harvest

Daily observation of the flower is required to determine when caps swell with resin and trichomes become more prominent, liquid accumulates and stand erect and sticky [49,51,52]. Maximum cannabinoid composition occurs when the trichomes’ colour changes into white, cloudy or milky, instead of the clear colour as shown in the Figure 2. The buds then produce high THC [49,51,53,54].

When the pastel hair colour turns to 75% light brown or amber, then inflorescences are ready for the harvest with a total CBD peak of 8.73% [55,56,57]. However, once the trichome starts looking grey and much of the THC has already degraded to CBN, the harvest time has passed, and the effects of the buds will be sleepy without any psychoactive effects [53,54,55]. Another method that has been used to determine the best time for the harvest of medicinal *Cannabis* is examining the physical appearance and budding mass by a digital microscope, 60× magnification via UV and LED light, magnifying glass and photographerloupe [49,54,58]. When the *Cannabis* is small, all the buds ripen simultaneously, but when the plant is larger, the first 3–6 inches of buds ripen before the inner buds [56]. In general, commercial harvest occurs after plants grow for 8–9 weeks, depending on the strain [52,56,59,60].

### 2.2. Harvesting Technology

The harvesting process happens when the medicinal *Cannabis* is in full flourishing [61,62,63]. In general, the manual process is the only way that has been used to harvest medicinal *Cannabis* without damage and produce high-grade flowers [64,65,66].

However, mechanical harvesting like rotary mowers and combined harvesting used mainly for high-stalk bast-fiber like hemp [64,67,68,69,70]. In general, a manual process is repeatable, low risk and maintains the crop’s quality [71].

Trimming the flower shortly after harvest is especially extensive for a high-grade whole flower of medicinal *Cannabis* [71]. All tools used during harvesting should be purified or sterilised [72]. There are four steps for trimming or manicuring ripe *Cannabis*, including clipping then cut-off the buds from the stem. After that, snip away the more minor, multi-fin leaves surrounding the buds [54,73]. The bud, after this step, should look naked and only a couple of the leaves stick between the flowers, and these leaves must be clipped off from the petiole parts [49]. The final steps before the drying are the wet trimming process, which includes removing the fan leaves, sugar leaves and any other extraneous parts of the plant, and collecting only manicured buds [49,74]. Different drying techniques have been applied for the drying of phytochemicals, and are summarised in Table 1 and further discussed in Section 3.

## 3. Drying of Cannabis

Many factors control the post-harvest quality. For instance, microbial activity, moisture content, room temperature, duration and light affect the quality and sustainability of medicinal *Cannabis* [45,46], so that few techniques have been developed to preserve the original phyto cannabinoid and terpenoid contents [43]. The most effective technique is drying, as *Cannabis* contains approximately 80% water. Benefits from drying include controlling microbial activity and enabling long-term storage while maintaining potency, taste and medicinal properties [75]. Most growers and commercial processors predicate the product is dry based on texture and crispness, while having only 11% *w/w* of moisture [76,91]. Any change in drying conditions may cause decarboxylation of acidic cannabinoids, loss of terpenes and reduced product quality [50]. Studies by ElSohly, Radwan, Gul, Chandra and Galal [44] and Taschwer and Schmid [45] found that the best way to avoid poor drying problems via a selection of a drying techniques depended on the strain’s chemical profile, drying behavior and the end product requirements. Several drying methods have been used to dry the flowers of *Cannabis*, including hot air drying, oven drying, vacuum freeze-drying, atmospheric freeze-drying and microwave-assisted drying. Drying techniques for medicinal *Cannabis* buds are summarised in Table 1.

### 3.1. Hot Air Drying or Hang Drying

Hang drying or hot air drying is one of the earliest drying methods [77], to dry either whole medicinal *Cannabis*, branches with flowers or trimmed flowers [75]. Firstly, remove the excess stems from the plants, transferring them to drying racks and hanging on either string lines, wire cages, or static wires upside-down to allow for air circulation and uniform drying by slowly transferring water from the stem, slowly migrating into the buds as the water evaporated [75,76]. Place the *Cannabis* material in a well-ventilated room. This room should be equipped with an environmental temperature control system set between 18–21 °C, relative humidity at 50–55% and air circulation using a small fan [76]. Under these controlled conditions, branches with flowers take 5–6 days to reach the desired moisture level, and trimmed flowers take only 4–5 days, but the whole plant takes 14 days to obtain 11% *w/w* moisture content [75]. Although this technique is mainly used, hot air drying takes a long time to process [76,77]. A study by Chandra, Lata, ElSohly, Walker and Potter [59] reported using a forced-air dryer for large scale drying. However, the disadvantage of this method is the requirements of additional curing time. Furthermore, the process requires daily inspection and proper maintenance of the conditions. Another disadvantage is the production of a product with a harsher taste due to the removing of buds from stems [92]. Another issue is often encountered with the traditional current method of drying is mould growth due to uncontrolled conditions [75]. Coffman and Gentner [78] evaluated the effect of drying conditions on the cannabinoid profile. It was found that the loss of the percentage of total cannabinoids and degradation of terpene increased from 7.5 to 11% when changing the process from 65 °C for 1 h to 105 °C for 64 h [76].

### 3.2. Oven Drying

Oven drying is one of the fast methods to dry medicinal *Cannabis* [87]. Many types of oven drying like a vacuum chamber, vacuum desiccator or drying oven with or without air circulation have been used to dry medicinal *Cannabis* [75]. The oven must be preheated at 37 °C for 24 h as optimal conditions to prevent decarboxylation for Phyto cannabinoids [79]. At any temperature higher than 37 °C, the sample contains both acid and neutral cannabinoids [79], and the total cannabinoids yield will decrease [78]. The oven had to be operated as a batch system, with buds hanging upside down. The benefit of this process is simple and low cost [93], and moisture accumulates in the flower until the capillaries begin to harden. Another advantage of using this method is ensuring the moisture will keep circulated inside the plant and prevent drying buds so quickly [75]. The disadvantage of this method is that it is only the best option for a small manufacturer of medicinal *Cannabis* [87]. This method will not cause any degradation or change of cannabinoids [79].

### 3.3. Microwave-Assisted Hot Air-Drying

Microwaves are non-ionising electromagnetic waves found between the radio and infrared wavelengths on the electromagnetic spectrum [94]. Microwave-assisted hot air drying is based on volumetric heating and creating a temperature gradient. The first stages of the drying process encourage the movement of the flow of moisture from the inner part to the surface [95,96], while the final stages of the drying process help with the removal of the bound water present in the material [97]. For most industrial applications, microwaves frequency of 915 MHz is considered the most valuable, due to its greater penetration depth. The significant advantages of microwave-assisted hot air drying are preserving nutrient contents, microstructure and colour of the dried sample [80,98]. The disadvantage is this process produces high heat, so it is only used in the final filling stage of the drying process. This high heat also causes an off taste [99]. Micro-wave-assisted hot air-drying preserves terpenes and cannabinoids [81,94]. A study by Kwaśnica, Pachura, Masztalerz, Figiel, Zimmer, Kupczyński, Wujcikowska, Carbonell-Barrachina, Szumny and Różański [82] found that drying with 240 W microwave-assisted hot air-drying cause maintained high quality 93 volatile compounds, predominantly β-myrcene, limonene and β-(E)-caryophyllene, as well as α-humulene similar to the chemical composition of fresh medicinal *Cannabis*.

### 3.4. Vacuum Freeze-Drying

The best method for drying medicinal *Cannabis* is freeze-drying the bud. Freeze-drying is carried out in three steps: freezing, primary drying and secondary drying [83,84]. During freezing, sample temperature is reduced to approximately −40 °C, thus converting most of the water present into ice. Freezing the product before drying is critical, as it determines the final quality. This step prevents the formation of water foam when the vacuum is applied [85]. In general, drying is complete when the sample’s temperature is the same as the system temperature [86]. After that, it must run secondary drying to remove residual moisture present in the sample without damaging proteins and lipids [83]. The preservation action is because of the structure of frozen materials, preventing the solid matrix’s degradation and resulting in a porous, unaltered structure [87].

In contrast, the glands and terpenes remain without damage. The main advantage of using vacuum freeze-drying in the *Cannabis* industry is that the low temperature inhibits microbial enzymatic activities and preserves the end product quality [100]. Tang and Pikal [83] show that secondary drying eliminates residual moisture. However, the main disadvantages of vacuum freeze-drying are the high initial capital and operational cost. It only permits drying in batches, and requires higher energy [101,102]. The best method for drying medicinal *Cannabis* is vacuum freeze-drying because it retains a maximal number of active compounds like the aroma and full flavour, and preserves the volatile compounds and acidic form of cannabinoids [84].

### 3.5. Microwave-Assisted Freeze Drying (MFD)

Another method used to dry medicinal *Cannabis* is microwave-assisted freeze drying. This process simply circulates cold, dry air over the frozen material at a temperature below −40 °C to −45 °C, pressure at 100 Pa, and microwave frequency of 2450 MHz [88,89,90]; this is to condense the vapors, maintain the frozen nature of the *Cannabis* and improve the mass transfer of water [101,102,103].

There are many types of atmospheric freeze-drying, such as a tunnel dryer, fluidised bed dryer and spray freeze dryer. Compared to tunnel dryers, the heat and mass transfer rates are better in fluidised bed dryers, while tunnel dryers can solve size reduction problems caused by mechanical shaking. The advantage of microwave freeze-drying (MFD) is 50–75% less time than freeze-drying [88]. Furthermore, Spray freeze-drying consists of a combination of spray drying and atomisation freeze-drying of the material, freezing and drying [103]. A study by Ishwarya, et al. [104] found that spray freeze-drying has higher volatile retention (93%) compared to freeze-drying (77%) and hot air spray drying (57%). However, there are many disadvantages, such as non-uniform heating of the dry zones in the product, impacting the product quality [105]. Its price is about four to ten times higher than other drying methods [89], and needs to be used in a limited amount to preserve high-quality hemp, and further research is required for testing and optimised CBD and terpenes contents [76].

## 4. Storage of Medicinal *Cannabis*

It is often impossible to immediately dry plants after harvest, but care must ensure that medicinal quality is retained during such holding periods. The first storage area is the holding area, where unprocessed buds before drying are stored, and the plants hang on string lines; whole plants can be hung upside down on taut rope or thin rods [49]. Removing the stem will save space, and the buds can be better held in boxes or trays [49]. The temperature must be controlled and kept at the 10–15 °C range, like a refrigerator or a small fan. *Cannabis* can be held for extended periods at low temperatures (4–6 °C), where the buds are not susceptible to mold growth [49,106]. Any late harvests or delays in the drying process will encourage mildew in wet buds due to significant amounts of more than 14% *w/w* moisture. Another reason is that *Cannabis* can also be contaminated during the harvest and post-harvest processing, exposing the medicinal plant to dangerous mycotoxins. Pests can flourish in storage. In general, after harvest, secondary metabolites start disrupted and chemical composition is altered [44,45]. Most microbial contamination occurs during the harvesting. Storing the product with poor drying conditions under high humid encourage the formation of microbial spores and harmful microbial toxins such as aflatoxin, as well as powdery mildew and botrytis, which is a cause of deterioration of medicinal *Cannabis* [59,107].

Drying and storage under uncontrolled conditions like moist and humid conditions can lead to fungal infections. Some mycotoxins like aflatoxins and ochratoxins require oxygen to grow. Therefore, eliminating or reducing the oxygen can retard the fungal growth during storage [108]. As well as poor drying and storing under humid conditions, this can promote microbial growth and toxins like aflatoxin and mycotoxin-producing strains of *Aspergillus* [109,110,111], as well as powdery mildew, *Botrytis* [106,112,113,114], *Cladosporium cladosporioides* [115], *Alternaria alternate* [112], *Verticillium* [106,116], *Salmonella* [117], *Enterobacter*, *Streptococcus* & *Klebsiella* [118]. However, the best solution is to store medicinal *Cannabis* at a water activity level below 0.3, and 11% *w/w* moisture content can activate microbial activity [59,107,113].

Another storage area is the curing-drying area, where the dried bud is stored until it is ready to be packaged. Firstly, the manicured buds should be naked by manually separating the bud from the stem or via EZ Trim de-budding. Secondly, bud trimming using a terminator on dry *Cannabis* [119]. After that, store the manicured buds in a cold, dry place at a temperature of 18 °C and relative humidity (RH) between 45–55%. This pure trimming offers many advantages, including maximising the cutting surface and faster trimming without damaging the flower. It also preserves the integrity of the trichome [49,119].

According to Caplan [120], the dried material should be cured at 18 °C and 60% RH for 14 days before determining the dry floral weight. As oxidation occurs with light, heat and oxygen, degradation of major cannabinoids is minimised after drying by storage in cool and dark places. Dried products must be stored between 1–5 °C, and frozen products must be kept at −18 °C to −20 °C for long-term storage [121]. The office of medicinal *Cannabis* of the Dutch government specifies that the water content of *Cannabis* must be between 5−10% directly after packing [122]. In commercial practice, dried *Cannabis* is stored for some time during the marketing of the bud or before further processing into therapeutic products. However, the published data on changes in chemical profiles during the storage of dried buds are only on changes in the neutral chemical profiles. Samples stored at 80°F were distinct from those stored at 120 °F; temperature significantly affects the chemical profiles. The profiles do not change over three months at the lower temperature. However, samples stored at 120 °F can be maximum up to 30 days without affecting the chemical profiles [123]. According to Taschwer and Schmid [45], samples can be stored at -18 °C or 4 °C for about 30 weeks before concentrations of THCA and THC change, although samples stored at 22 °C ± 1 °C showed some rapid decomposition. Even though *Cannabis* buds stored at 100 °C and 150 °C showed significant decarboxylation of THCA and decomposition of THC within two hours, dried samples stored at 50 °C for 24 h showed slight decarboxylation. Milay, Berman, Shapira, Guberman and Meiri [43] generally claim that the most unfavourable temperature for storage is 25 °C, as it causes enormous changes in natural Phyto cannabinoids over time, compared to 4, −30 and −80 °C.

Packaging material should not be plastic bags, as they encourage mould growth–instead use only a glass jar. When storing the bud with infected powdery mildew, mould depletes the oxygen in the jar. Then herbal fragrances are replaced with the astringent odour of ammonia, and the plant turns brown and crumbly [49]. When the moist bud is packed in a sealed container in warm conditions, mould may not germinate; instead, bacteria become active. First, aerobic bacteria ingest tissue when they deplete the oxygen; the anaerobic bacteria start feasting and releasing ammonia [49]. Whenever the bud smells bad, repack products and do the drying tests. The best ways to store the buds are either glass, ceramic, metal or wood containers, as no electrical charge that attracts the glands are airtight and retain the bud dry. Maximum storing time for the dry bud is up to one year in the freezer, with virtually no deterioration [72]. The finished product must contain active ingredients like CBD and THC that comply with the qualitative and quantitative composition of good manufacture product (GMP) guide [72,124,125].

## 5. Conclusions

The Cannabis industry has been flourishing in recent years due to its legalisation and depenalisation as a medicinal product by many countries around the world. Its therapeutic potential is dependent a lot on the initial quality of the Cannabis plant material, as influenced by the harvesting time and harvesting technique and its post-harvesting technologies, which determine the chemical composition and quality of the products. Safety has been the biggest concern, however, due to the fact that no proper standardised procedures are present. Thus, having controlled processing systems in place in alignment with the scientific evidence is essential. In addition, most of these post-harvesting techniques, especially drying, have been trialed in small scale operations. How these small-scale operations translate into large-scale ones is still a question, and a challenge. The exact replication of ideal drying conditions on large-scale operations with no shortcuts needs to be engineered precisely within the equipment setup. Some challenges include space availability for drying rooms, uneven drying, growth of mould based on drying speeds, environmental controls (humidity and temperature), overhead costs such as labour and construction and loss of aroma and flavours. Some of these challenges can be overcome by having multiple drying areas or bins that can constantly tune the exact conditions based on the amount, maturity and type of *Cannabis*. New advances such as fully automated operations should be trialed to *Cannabis* productions to produce good quality, safe products with increased process efficiency in a sustainable manner. In conclusion, the most suitable drying methods used in the *Cannabis* industry are hot air drying or hang drying, oven drying, microwave-assisted hot air-drying, vacuum freeze-drying and microwave-assisted freeze drying (MFD). Vacuum freeze-dry is considered the best method to retain a maximal number of active phytochemicals, preserve the quality of terpenes, and is a safe method to develop medicinal *Cannabis* products that takes only a short time. Furthermore, the impacts of the variations related to operation production cost and *Cannabis* type (strain, chemotypes and chemovars) on the preservation of terpenes, cannabinoids and phenolic compounds in dried buds should also be considered when designing future studies.

## Figures and Tables

**Figure 1 molecules-27-01719-f001:**
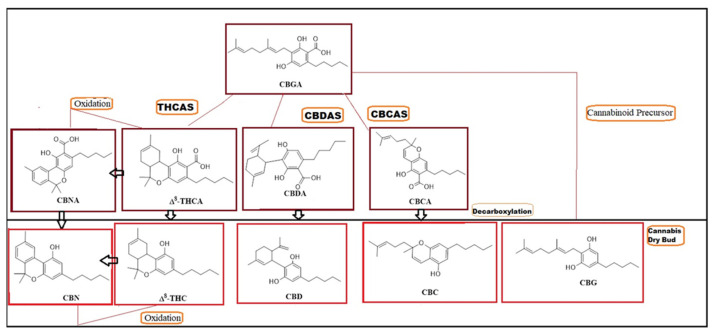
Structures of major components in the medicinal *Cannabis*. cannabigerolic acid, CBGA; tetrahydrocannabinolic acid synthase, THCAS; cannabidiolic acid synthase, CBDAS; cannabichromene acid synthase, CBCAS; tetrahydrocannabinolic acid, THCA; cannabidiolic acid, CBDA; cannabichromenic acid, CBCA; cannabinolic acid, CBNA; cannabichromene, CBC; cannabidiol, CBD; cannabigerol, CBG; cannabinol, CBN; and tetrahydrocannabinol, THC.

**Figure 2 molecules-27-01719-f002:**
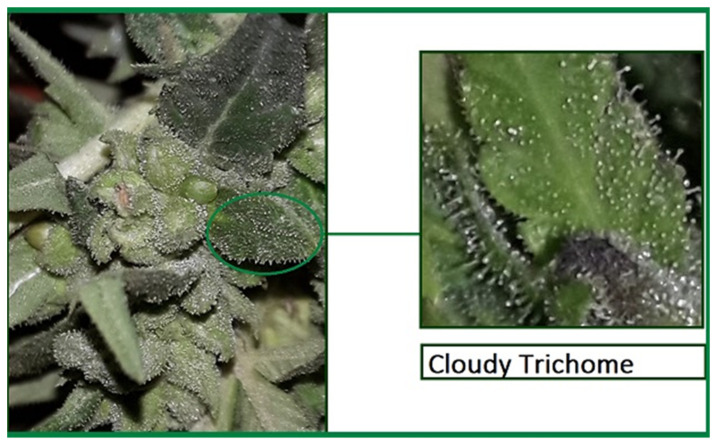
Morphological appearance of white or milky trichome of medicinal *Cannabis.*

**Table 1 molecules-27-01719-t001:** Drying techniques for medicinal *Cannabis* buds.

Drying Technique	Drying Conditions/Procedures	Advantages and Disadvantage	References
Hot Air Drying	The plant materials were hanged on either string lines, wire cages, or static wires upside-down to allow for air circulation and uniform drying by control system has been set between 18–21 °C, relative humidity at 50–55% and air circulation using a small fan under these controlled conditions. Trimmed flowers take only 4–5 days, but the whole plant takes up to 14 days.	A simple technique, but required regularly maintain optimal conditions.	[59,75,76,77,78]
Oven Drying	Buds were hanging upside down in the oven and oven must be preheated at 37 °C for 24 h to prevent decarboxylation for Phyto cannabinoids	A simple technique, but under optimal conditions and difficult for commercial production.	[75,78,79]
Microwave-assisted hot air-drying	Samples were dried by applied volumetric heating and creating a temperature gradient and standard microwaves frequency set at 915 MHz and 240 W to maintain high-quality medicinal cannabis	An advanced technique, but under optimal conditions.	[80,81,82]
Vacuum Freeze-Drying	Vacuum freezing the cannabis bud by reducing the temperature to approximately −40 °C before drying the buds to retain a high quality of phytochemicals.	Quite effective and most suitable advanced technique, but prohibitive operational cost.	[83,84,85,86,87]
Microwave-Assisted Freeze Drying	Circulates cold, dry air over the frozen material at a temperature below −40 °C to −45 °C, pressure at 100 Pa, microwave frequency 2450 MHz.	An advanced technique, but under optimal conditions.	[76,88,89,90]

## Data Availability

Not applicable.

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
