# Peer review of "Post-Harvest Operations to Generate High-Quality Medicinal Cannabis Products: A Systemic Review"

_molecules, 2022, doi:10.3390/molecules27051719_

Round 1

Reviewer 1 Report

In an article by Hebah M.S. AL Ubeed "Postharvest operations to generate high quality medicinal cannabis products: a systemic review" presents the impact of preparation of plant material on the quality of the crop. The article is very interesting and introduces new elements. Cannabis plant is a very interesting plant that is more and more appreciated.

Minor errors:

1) underline the novelty element in intriduction.

2) the abstract lacks the specified purpose of the work.

3) proposes to include some structural formulas of the most important active compounds found in this plant.

Recommend publication in Molecules MDPI.

Author Response

  • Underline the novelty element in introduction.

Response: Thanks for the suggestions, amended as: “Postharvest reports optimising the drying techniques and factors to recover bioactive compounds such as terpenes, cannabinoids, and phenolic compounds from the medicinal marijuana are limited in the literature so that there is a need to understand the postharvest handling of Cannabis that optimize yield and control high-quality medical-grade product. Typical processing of Cannabis involves harvesting at the optimal time, drying the harvested product with minimal loss of active constituents, extracting cannabinoids, then storage the product for further process. This review portrays the knowledge for the optimal benefit from the postharvest process is standardized to produce therapeutically stable Cannabis products for patients by a deeper understanding of the Cannabis chemical space, outlining major phytochemicals in Cannabis, reviewing the current drying techniques and optimum methods which have been applied for Cannabis-derived phytochemicals, highly efficient mechanical drying methods are needed, and how to effectively reduce the costs of drying and phytochemical isolation is crucial for further applications. To date, there are not many papers that discuss the post-harvesting strategies for medicinal Cannabis and managing the quality of export Cannabis.”   

  • The abstract lacks the specified purpose of the work.

Response: Amended as directed, thanks

“The traditional Cannabis plant as a medicinal crop has been explored for many thousand years. The Cannabis industry is rapidly growing; therefore, optimising drying methods and producing high-quality medical products have been a hot topic in recent years. We systemically analysed the current literature and drew a critical summary of the drying methods implemented thus far to preserve the quality of bioactive compounds from medicinal Cannabis. Different drying techniques have been one of the focal points during the post harvesting operations as drying preserves these Cannabis products with increased shelf life. Followed or even highlighted the most popular methods used. Drying methods have advanced from traditional hot air and oven drying methods to microwave-assisted hot air drying or freeze-drying. In this review, traditional and modern drying technologies are reviewed. Each technology will have different pros and cons of its own. Moreover, this review outlines the quality of the Cannabis plant component harvested plays a major role in drying efficiency and preserving the chemical constituents. The emerge of medical Cannabis, and cannabinoid research requires optimal post-harvesting processes for different Cannabis strains. We proposed the most suitable method for drying medicinal Cannabis to produce consistent, reliable, and potent medicinal Cannabis. In addition, drying temperature, rate of drying, mode, and storage conditions after drying influenced the cannabis component retention and quality.”

  • Proposes to include some structural formulas of the most important active compounds found in this plant.

Response: Thanks for the suggestion; we now have added figure 1 consisting of structures of major components in the medicinal Cannabis.

The figure.1 has been referenced in the text in Line 60.

Structural formulas of the most important active compounds found in this plant was also well-reviewed by a recent review in Molecules by Radwan et al. 2021 (Radwan, M.M.; Chandra, S.; Gul, S.; ElSohly, M.A. Cannabinoids, phenolics, terpenes and alkaloids of cannabis. Molecules 2021, 26, 2774.), which we also cited in our current manuscript.

Reviewer 2 Report

This manuscript reviewed the post-harvest operations for cannabis products, from harvesting to storage. Due to the legalization and depenalization of medical/recreational marijuana in many more countries, cannabinoids like THC and CBD have attracted much more attention in the field. Such a review will enhance the understanding of critical processes of cannabis production and attract attention from medicinal chemists and analytical chemists in cannabinoid research.

Hence, this manuscript agrees with the special issue and can be accepted to publish in Molecules after minor revisions.

  • A figure with structures of major components in cannabis preparation would enhance the value of this manuscript. The figure could include the chemical structures of THC, THCV, CBD, CBC, CBG, CBN, and their corresponding acids, etc. 

  • A table summary of drying techniques with their advantages and disadvantages would help the viewers compare and understand the differences in various drying techniques. This will also enhance the value of this manuscript.  

Author Response

  • A figure with structures of major components in cannabis preparation would enhance the value of this manuscript. The figure could include the chemical structures of THC, THCV, CBD, CBC, CBG, CBN, and their corresponding acids, etc. 

Response: Thanks for the suggestion; we now have added figure 1 consisting of structures of major components in the medicinal Cannabis.

The figure.1 has been referenced in the text in Line 60

Structural formulas of the most important active compounds found in this plant was also well-reviewed by a recent review in Molecules by Radwan et al. 2021 (Radwan, M.M.; Chandra, S.; Gul, S.; ElSohly, M.A. Cannabinoids, phenolics, terpenes and alkaloids of cannabis. Molecules 2021, 26, 2774.), which we also cited in our current manuscript.

  • A table summary of drying techniques with their advantages and disadvantages would help the viewers compare and understand the differences in various drying techniques. This will also enhance the value of this manuscript.  

Response: We would like to thank the reviewer for the constructive suggestion. We have added a new Table (Table 1), which summarized the drying techniques, their procedures as well as their advantages and disadvantage in the revised manuscript to provide comprehensive information to the readers. 

Reviewer 3 Report

This manuscript under the title "Postharvest operations to generate high-quality medicinal Cannabis products: a systemic review", include interesting information about Cannabis and how to handle it properly during harvesting, drying, and storage without affecting their active constituents and accordingly its activity.

However, it presented most of the methods followed but they didn't come to a conclusion what are the most suitable and best ways to be followed or even highlight the most popular methods used.

Also, the advantages and disadvantages of each method should be included. In simple when handling Cannabis what are the optimum methods from all these presented and why?

The manuscript should include the author's point of view and suggestions after the collection of all these data and reading in this field.

In the conclusion section line 357 it is stated that " these small scale operations translate into large scale operations is still a question and a challenge". What are your suggestions to pass this challenge, difficulties, advantages, disadvantages, facilities that should be available, precautions that should be considered.

Author Response

However, it presented most of the methods followed but they didn't come to a conclusion what are the most suitable and best ways to be followed or even highlight the most popular methods used.

Response: Amended as directed, thanks.

In conclusion, the most suitable drying methods used in the cannabis industry are hot air drying or hang drying, oven drying, Microwave-assisted hot air-drying, Vacuum Freeze-Drying and Microwave-Assisted Freeze Drying (MFD). Vacuum freeze-dry is considered the best method to retain a maximal number of active phytochemicals, preserve the quality of terpenes, a safe method to develop medicinal cannabis products and take only a short time.

Also, the advantages and disadvantages of each method should be included. In simple when handling Cannabis what are the optimum methods from all these presented and why?

Response: We would like to thank the reviewer for the constructive suggestion. We have added a new Table (Table 1), which summarized the drying techniques, their procedures as well as their advantages and disadvantage in the revised manuscript to provide comprehensive information to the readers. 

The manuscript should include the author's point of view and suggestions after the collection of all these data and reading in this field.

Response: Amended as directed, thanks.

Furthermore, the impacts of the variations related to operation production cost, Cannabis type (strain, chemotypes, and chemovars), on the preservation of terpenes, cannabinoids, and phenolic compounds in dried buds should also be considered whilst designing future studies.

In the conclusion section line 357 it is stated that " these small scale operations translate into large scale operations is still a question and a challenge". What are your suggestions to pass this challenge, difficulties, advantages, disadvantages, facilities that should be available, precautions that should be considered.

Response: Thanks for the constructive suggestion and the conclusion has been amended as follows to reflect the reviewer’s suggestion” How these small-scale operations translate into large scale operations is still a question and a challenge. Exact replication of ideal drying conditions on large scale operations with no shortcuts need to be engineered precisely within the equipment set up. Some challenges include space availability for drying rooms, uneven drying, growth of molds based on drying speeds, environmental controls (humidity and temperature), overhead costs such as labor and construction and loss of aroma and flavour. Some of these challenges can be overcome by having multiple drying areas or bins that can constantly tune the exact conditions based on the amount, maturity, and type of cannabis. New advances such as fully automated operations should be trialed to Cannabis productions to produce good quality, safe products with increased process efficiency in a sustainable manner.

Round 2

Reviewer 3 Report

Accept in current form